# Statin Treatment for Reducing Mortality Risk in Individuals over 75 Years of Age: A Large-Scale Retrospective Analysis

**DOI:** 10.3390/jcm14165739

**Published:** 2025-08-14

**Authors:** Noy Nachmias, Sher Matsri, Maisaa Sharary, Noam Yaniv, Tal Netser, Assaf Buch, Yona Greenman, Elena Izkhakov, Eugene Feigin

**Affiliations:** 1Institute of Gastroenterology and Liver Diseases, Tel Aviv Sourasky Medical Center, Faculty of Medical and Health Sciences, Tel Aviv University, Tel Aviv 69978, Israel; 2Faculty of Medicine, Tel Aviv University, Institute of Endocrinology, Metabolism and Hypertension, Tel Aviv Sourasky Medical Center, Tel Aviv 69978, Israel; 3Department of Nutrition Sciences, School of Health Sciences, Ariel University, Ariel 40700, Israel

**Keywords:** statins, elderly, all-cause mortality

## Abstract

**Background**: Despite the worldwide increase in life expectancy, individuals aged 75 years and older with an unknown history of cardiovascular disease often receive suboptimal statin treatment for primary prevention, reflecting uncertainties regarding statin efficacy and safety in this aging group. We aimed to assess the impact of statin treatment on all-cause mortality among individuals aged 75 years and older without prior cardiovascular diagnoses. **Methods**: This retrospective study utilized real-world data from a large cohort of individuals aged 75 years and older who were treated as outpatients in or were admitted to the Tel Aviv Sourasky Medical Center. Extracted variables included demographic details, Charlson Comorbidity Index (CCI), chronic medication regimens, mortality outcomes and blood test results (high-density lipoprotein, low-density lipoprotein and creatinine). Patients with a prior diagnosis of angina, myocardial infarction or stroke were excluded from the study. **Results**: A total of 98,502 patients were included in the study, of whom 37,171 (mean age 80.67 ± 4.73 years) were treated with statins and 6804 (18.3%) of the latter patients were aged 85 years and above. The majority of the statin-treated patients (72.6%) had received high-intensity statins. The non-statin-treated group comprised 61,331 subjects with a mean age of 82.69 ± 5.77 years, of whom 19,253 (31.39%) were aged 85 years and above. The risk of all-cause mortality was significantly lower in the statin-treated group compared to the non-statin-treated group (adjusted odds ratio [aOR] 0.715, 95% confidence interval [CI] [0.671–0.761], *p* < 0.001). This trend persisted after stratification for age 85 years and above (aOR 0.7, 95% CI [0.606–0.809], *p* < 0.001), and for a low CCI (≤4) or a high CCI (>4) score (aOR 0.766, 95% CI [0.708–0.803]; aOR 0.648, 95% CI [0.585–0.717], respectively, *p* < 0.001). **Conclusions**: Provision of statin therapy contributes to a reduction in risk of all-cause mortality in individuals aged 75 years and above who have an unknown history of cardiovascular disease, regardless of the type of statin or the patient’s CCI score.

## 1. Background

The aged population, particularly the population of the oldest old (individuals over 80 or 85 years of age), is predicted to multiply worldwide in the coming decades [1]. The absolute risk of cardiovascular disease increases together with an increase in the number of hospitalizations of patients above the age of 75 years due to acute coronary syndrome [2]. Statin treatment is considered a cardinal drug in reducing the risk for atherosclerotic cardiovascular disease (ASCVD) by affording both primary and secondary prevention [3,4]. However, absolute recommendations for primary prevention above the age of 75 years are lacking but rather based upon individual risk assessment [3,4].

Data on primary prevention of ASCVD with statins are scarce in this population. Patients older than 75 years of age tend to be underrepresented in clinical trials and undertreated with statins for primary prevention [2,5]. This is in spite of several studies that showed that adequate low-density lipoprotein (LDL) cholesterol control can reduce cardiovascular events in patients older than 75 years and should be considered as chronic treatment for this population [6,7]. The Copenhagen general population study included 91,161 non-diabetic patients with no history of cardiovascular disease and no previous treatment with statins. That study’s results showed that the risk for coronary events for patients aged between 70 and 100 years was higher compared to younger adults (50–70 years) [6]. The number needed to treat (NNT) to prevent one cardiovascular event in 5 years was lower for older subjects (>80 years, *n* = 42) than for younger subjects (50–59 years, *n* = 439) [6]. A similar trend of favorable results using lipid-lowering therapy in the elderly population was observed in a meta-analysis by Gencer et al., which included 21,442 patients above the age of 75 years, most of them with a known cardiovascular disease treated with statins, ezetimibe and PCSK9 [7]. Other studies focusing upon similar populations showed mixed and even contradictory results regarding cardiovascular outcomes in geriatric patients treated with any form of statin therapy [6,8].

The aim of the current study was to investigate the impact of statin treatment on all-cause mortality among an advanced-age population (75–85 years) as well as an oldest old population (above 85 years), both without a known cardiovascular disease.

## 2. Methods

This retrospective cohort study was approved by the Tel Aviv Sourasky Medical Center (TASMC) institutional review board (IRB) (TLV-0459-23). Informed consent was waived by the IRB according to local policy for anonymized retrospective studies. We followed the Strengthening the Reporting of Observational Studies in Epidemiology (STROBE) reporting guidelines [9].

### 2.1. Study Population

This real-world data retrospective cohort study included all medical records of the TASMC, a tertiary, university-affiliated 1170-bed acute care hospital, since the beginning of electronic medical record use in June 2003 until April 2023. The study population included all patients aged 75 years and older who were admitted to any of the hospital’s departments, outpatient clinics and emergency rooms. The records of patients with a prior diagnosis of myocardial infarction (MI) or cerebrovascular accidents (CVAs) were excluded from the study.

### 2.2. Data Source and Variables

Data were obtained by means of MDClone, a query tool that provides comprehensive patient-level data on wide-ranging variables in a defined timeframe around an index event [10]. The following data were collected for each patient, based on documented medical history: age; sex; chronic drug treatment, specifically aspirin and statin use; mortality; blood test results (high-density lipoprotein [HDL], LDL, and creatinine); and calculation of the Charlson Comorbidity Index (CCI) [11]. To ensure the exclusion of patients with a known history of MI or CVA, we applied two strategies: First, we excluded all patients with a medical history of the following diagnoses: acute MI including ST elevation and non-ST elevation MIs (International Classification of Diseases, Ninth Revision, Clinical Modification (ICD-9-CM) codes 410, 410.01, 410.02, 410.1, 410.11, 410.12, 410.2, 410.21, 410.22, 410.3, 410.31, 410.32, 410.4, 410.41, 410.42, 410.5, 410.51, 410.52, 410.6, 410.61,410.62, 410.7, 410.71, 410.72, 410.8, 410.81, 410.82, 410.9, 410.91, 410.92), unstable angina (ICD-9-CM code 411.1) and other and unspecified angina pectoris (ICD-9-CM code 413.9). Also excluded were patients who had undergone surgery on vessels of the heart, such as the removal of coronary artery obstruction and insertion of a stent (ICD-9-CM code 36.0), bypass anastomoses for heart revascularization (ICD-9-CM code 36.1), heart revascularizations by heart implant (ICD-9-CM code 36.2), other heart revascularization procedures (ICD-9-CM code 36.3), other operations on vessels of the heart (ICD-9-CM code 36.9), a CVA including ischemic stroke (ICD-9-CM codes 433, 434, 436) and hemorrhagic stroke (ICD-9-CM codes 430–432) and a transient ischemic attack (ICD-9-CM code 435.9). Second, the CCI was further used to exclude patients with comorbid conditions indicative of cardiovascular disease, including MI, chronic heart failure, peripheral vascular disease and history of CVA or TIA [11].

For each patient, if deceased, we collected the date of death as documented in the internal ministry or as a diagnosis of Exitus (ICD-9-CM code 798.1 and 798.2) documented in the medical records of Tel Aviv Sourasky Medical Center.

### 2.3. Study Endpoints

The primary endpoint was 1-year and 10-year all-cause mortality among individuals aged 75 years and older with and without chronic statin treatment. The secondary endpoint was all-cause mortality among the oldest old population aged 85 years and older with and without chronic statin treatment.

### 2.4. Statistical Analysis

Categorical variables were compared using the Chi-squared test, and continuous variables were compared using the Mann–Whitney U test. To evaluate the association between statin use and outcomes (all-cause mortality, MI and CVA), we performed multivariable logistic regression for 1-year outcomes and Cox proportional hazards models for long-term outcomes (CVA and MI). The models were adjusted for relevant covariates including demographics (age, sex), CCI, concomitant chronic aspirin treatment and HDL, LDL, and creatinine levels and then stratified to high- vs. moderate- and low-intensity statins. Propensity score methods (e.g., matching or inverse probability weighting) were considered; however, multivariable adjustment was selected due to the large sample size and to retain all available data in the analysis [12]. Model diagnostics, including multicollinearity and proportional hazard assumptions, were assessed and met. All statistical tests were two-tailed, and *p* < 0.05 was considered statistically significant. All statistical analyses were performed using SPSS (IBM SPSS Statistics for Windows, version 27, IBM Corp., Armonk, NY, USA, 2020).

## 3. Results

### 3.1. Baseline Characteristics

Table 1 summarizes the baseline characteristics of the cohort. Between June 2004 and April 2023, a total of 106,856 patients aged 75 years or older were admitted to TASMC. Of these, 98,502 patients without a history of ischemic vascular events (MI or CVA) were included in the study.

The **non-statin group** consisted of 61,331 patients (57.0% female), with a mean age of 82.69 ± 5.77 years. The **statin-treated group** included 37,171 patients (52.2% female), with a significantly lower mean age of 80.62 ± 4.73 years (*p* < 0.001).

Patients aged 75–85 years accounted for 68.6% of the non-statin group (42,078 patients) and 81.7% of the statin group (30,367 patients), while those older than 85 years represented a higher proportion in the non-statin group (31.4% vs. 18.3%; *p* < 0.001).

The incidence of hypertension was significantly higher in the statin group compared to the non-statin group (77.9% vs. 44.0%; *p* < 0.001). Similarly, aspirin use was more common among statin users (37.3% vs. 8.0%; *p* < 0.001).

Mean creatinine levels were slightly higher in the statin group (1.22 ± 0.75 vs. 1.03 ± 0.79 mg/dL), while HDL levels were marginally lower (45.4 ± 15.58 vs. 46.9 ± 16.93 mg/dL). LDL levels were also lower in the statin group (83.5 ± 34.6 vs. 94.0 ± 34.7 mg/dL). The median Charlson Comorbidity Index (CCI) was 4 (IQR 3–5) in both groups.

Among those receiving statins, 27.4% (10,186 patients) were on high-intensity regimens, while 72.6% (26,985 patients) received low- or moderate-intensity statins. A small subset (1.7%) were prescribed a combination of statins and ezetimibe.

### 3.2. CVA, MI and All-Cause Mortality

**In the non-statin group**, 1-year all-cause mortality was 23.04% (14,131 patients), and 10-year all-cause mortality was 69.8% (42,805 patients). Over the 10-year follow-up, 5.58% (3426 patients) experienced a CVA, and 5.62% (3450 patients) experienced an MI.

In comparison, the **statin group** had significantly lower mortality rates: 1-year all-cause mortality was 10.24% (3805 patients), and 10-year mortality was 51.85% (19,274 patients) (*p* < 0.001 for both).

However, the incidence of CVA and MI was significantly higher **in the statin group**, with 16.88% (6277 patients) experiencing a CVA and 17.79% (6613 patients) an MI (*p* < 0.001 for both comparisons vs. non-statin group) (Table 2).

### 3.3. Impact of Statin Intensity on All-Cause Mortality

Chronic treatment with statins of any potency (low/medium- or high-intensity) was associated with lower all-cause mortality (OR 0.466, 95% CI [0.454–0.479], *p* < 0.001, aOR 0.715, 95% CI [0.671–761], *p* < 0.001) (Figure 1). The patients were stratified into three groups of low/moderate-intensity, high-intensity, and no statin treatment, with the following findings: The low/moderate-intensity statin group had an OR of 0.521, 95% CI [0.506–0.537], *p* < 0.001, and an aOR of 0.801, 95% CI [0.749–0.857], *p* < 0.001. The high-intensity statin group had an OR of 0.347 95% CI [0.333–0.365], and an aOR of 0.567, 95% CI [0.523–0.615], *p* < 0.001 (Figure 1).

### 3.4. Impact of CCI on All-Cause Mortality

After stratifying the cohort into two CCI groups of ≤4 and >4, the use of statins was correlated with lower mortality in the low- and high-CCI groups (aOR 0.766, 95% CI [ 0.708–0.803], *p* < 0.001 and aOR 0.648, 95% CI [0.585–0.717], *p* < 0.001, respectively). After additionally stratifying the cohort to low- and moderate-intensity statin vs. high-intensity statins, the low-CCI group’s low- and moderate-intensity statins had an OR of 0.592, 95% CI [0.571–0.614], *p* < 0.001, and an aOR of 0.868, 95% CI [0.797–0.945], *p* = 0.001; its high-intensity statins had an OR of 0.381, 95% CI [0.360–0.402], *p* < 0.001, and an aOR of 0.579, 95% CI [0.521–0.644], *p* < 0.001. The high-CCI group’s low- and moderate-intensity statins had an OR of 0.506, 95% CI [0.478–0.535], *p* < 0.001, and an aOR of 0.704, 95% CI [0.63–0.787], *p* < 0.001, and its high-intensity statins had an OR 0.337, 95% CI [0.314–0.362], *p* < 0.001, and an aOR of 0.559, 95% CI [0.492–0.636], *p* < 0.001.

### 3.5. Impact of Statin Intensity on Cardiovascular Outcomes

Chronic treatment with statins of any kind (low/medium- or high-intensity) was associated with a higher incidence of MIs, with an OR of 3.631, 95% CI [3.476–3.792], *p* < 0.001 and an aOR of 1.799, 95% CI [1.68–1.926]. It was also associated with a higher incidence of CVA events, with an OR of 3.434, 95% CI [3.287–3.588], *p* < 0.001 and an aOR of 1.798, 95% CI [1.671–1.934], *p* < 0.001.

### 3.6. Oldest Old Population

Any statin treatment in the oldest old population (i.e., aged over 85 years) was associated with lower mortality rates with an OR of 0.489, 95% CI [0.460–0.52], *p* < 0.001 and an aOR of 0.7, 95% CI [0.606–0.809], *p* < 0.001. Low- and moderate-intensity statins had an OR of 0.552, 95% CI [0.515–0.592], *p* < 0.001 and an aOR of 0.828, 95% CI [0.708–0.970], *p* = 0.019, while high-intensity statins had an OR of 0.331, 95% CI [0.297–0.369], *p* < 0.001 and an aOR of 0.47, 95% CI [0.387–0.579], *p* < 0.001.

## 4. Discussion

The current approach to prophylactic statin treatment among the aged population remains suboptimal [13], with existing guidelines primarily targeting adults up to the age of 75 [14]. This retrospective study aimed to evaluate the impact of chronic statin therapy on all-cause mortality in individuals aged 75 and older without a known history of cardiovascular disease. Our findings demonstrated a consistent protective effect of chronic statin treatment on all-cause mortality across all sub-group analyzed. Notably, treatment with high-intensity statins was found to be even more protective. The greater-risk patients, such as those with a high CCI or those of older age, had greater benefit than patients with a lower CCI as well as those who were younger, and the risk of CVD was lower among the statin-treated group during the 10-year follow-up. There was a difference between low/moderate- and high-intensity statins in favor of the latter among all age and CCI groups, further advocating for statin treatment rather than a more conservative approach.

Interestingly, we observed a paradoxical finding: while all-cause mortality was lower in the statin-treated group, the incidence of CVA and MI was significantly higher (1.83 and 1.92 times, respectively). This finding is consistent with findings from previous studies involving high-risk populations [8,13]. Of note, this discrepancy may reflect several methodological and clinical factors. First, patients selected for statin therapy are likely to have had higher baseline cardiovascular risk, as reflected in our cohort by the higher prevalence of hypertension and lower HDL levels in the statin group [4,14]. These factors may have predisposed them to more events despite statin treatment.

Second, surveillance bias may have contributed—patients on statins are often more closely monitored, increasing the likelihood of capturing non-fatal events [15]. Third, the longer survival time among statin users may have allowed more time for cardiovascular events to occur, whereas individuals in the non-statin group may have died earlier from other causes.

Additionally, differences in data sources and event definitions could contribute. Mortality data were obtained from national registries, ensuring complete capture of deaths regardless of setting. In contrast, MI and CVA data were based solely on events recorded at our institution. MI was defined broadly, including all troponin elevations at admission, which may have captured non-ischemic type 2 MIs—often seen in elderly or acutely ill patients and not typically preventable by statins. CVA was based on new stroke or TIA diagnoses during follow-up at our hospital; events treated elsewhere or diagnosed without hospital contact may have been missed. Moreover, in-hospital deaths were sometimes recorded generically as “Exitus,” without identifying the specific cause, leading to possible underreporting of fatal cardiovascular events.

Given these limitations, we were unable to construct a reliable composite endpoint such as MACEs (major adverse cardiovascular events), which could have more accurately reflected cardiovascular risk [16].

Earlier studies show mixed results on statins for primary prevention in older adults [13,17]. Meta-analyses by the Cholesterol Treatment Trialists’ Collaboration have shown that while statins reduce major vascular events, the absolute benefit decreases with age, particularly in those over 75 years without prior cardiovascular disease [17]. The ongoing STAREE trial (Statins for the Extension of Disability-Free Life and Primary Prevention of Cardiovascular Events in the Elderly) will provide prospective, high-quality data to better define the utility of statins in this population [18,19].

We incorporated the Charlson Comorbidity Index (CCI) to reflect the burden of comorbid conditions in the elderly, since traditional risk calculators like ASCVD are not validated for individuals over 75. A CCI ≥ 4 has been associated with increased mortality risk [11,20]. Interestingly, our analysis showed that statin use was associated with reduced mortality even among patients with high CCI scores, suggesting that comorbidity burden should not automatically preclude statin initiation. Instead, the CCI may be a valuable tool in individualizing decisions regarding statin therapy in older patients.

### Strengths and Limitations

Strengths of our study include the large, real-world cohort of patients aged ≥ 75 years, comprehensive mortality data from national registries, and adjustment for key clinical variables including age, sex, creatinine, lipid levels, hypertension, aspirin use, and CCI.

However, several limitations should be acknowledged. First, the retrospective design introduces the risk of residual confounding and selection bias. While we adjusted for key baseline variables using multivariable logistic regression, there were notable differences between groups—particularly in age, hypertension, and aspirin use. Propensity score matching or inverse probability weighting could have improved group comparability but were not performed to maintain full sample size and statistical power [12]. This choice may limit the extent to which confounding can be controlled.

Second, although we excluded patients with documented ASCVD (MI, CVA/TIA, CHF, peripheral vascular disease) [14], undiagnosed or undocumented disease (e.g., aortic atherosclerosis or asymptomatic carotid disease) may have been present and could not be completely ruled out. Similarly, atrial fibrillation and diagnoses of arrhythmia—a key contributor to stroke risk—were not evaluated due to limitations in data extraction [21,22].

Third, we lacked data on important lifestyle factors such as smoking, diet, physical activity and alcohol use, which may confound the relationship between statin use and outcomes. Additionally, we did not assess the side effects of statins, particularly neuromuscular symptoms [23], polypharmacy and drug–drug interactions, that may affect adherence in older adults [24].

Lastly, while our mortality data are comprehensive, event data (MI and CVA) were limited to our institution. As such, events that occurred elsewhere or went undocumented may have led to underestimation.

## 5. Conclusions

The findings of our study demonstrated that the elderly population aged ≥ 75 years, with an unknown history of CVD, might benefit from statin treatment due to reduced risk of all-cause mortality. The risk reduction was independent of statin type and CCI score.

## Figures and Tables

**Figure 1 jcm-14-05739-f001:**
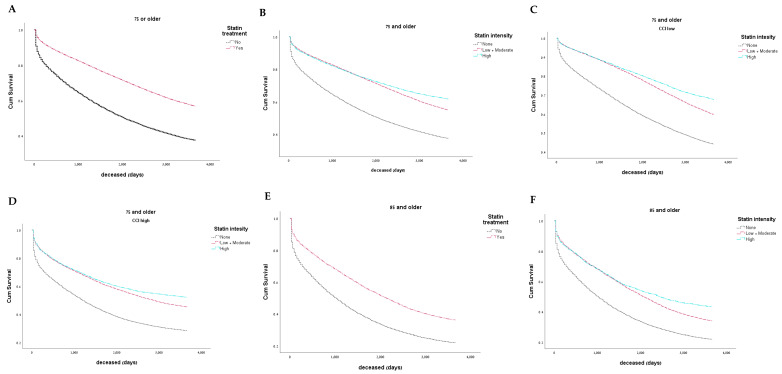
(**A**)—Survival of the statin-treated group compared to the statin-naïve group. (**B**)—Survival of the statin-treated group stratified to high-, low- or moderate-intensity statins compared to the naive group. (**C**)—Survival among patients with a CCI ≤ 4 in the statin-treated group compared to the statin-naive group. (**D**)—Survival among patients with a CCI > 4 in the statin-treated group compared to the statin-naive group. (**E**)—Survival among the oldest old (aged over 85 years), statin-treated group compared to the statin-naive group. (**F**)—Survival among the oldest old (aged over 85 years) statin-treated group stratified to high-, low- or moderate-intensity statins compared to the statin-naïve group. CCI—Charlson Comorbidity Index.

**Table 1 jcm-14-05739-t001:** Characteristics of cohort and comparison of both groups with and without chronic statin treatment.

	Without Chronic Statin Treatment (N = 61331)		With Chronic Statin Treatment (N = 37171)		
**Male**	26,388 (43%)		17,772 (47.8%)		
**Mean Age**	82.69	±5.77	80.67	±4.73	*p* < 0.001
**75 to 85 years**	42,078	68.61%	30,367	81.7%	*p* < 0.001
**85+**	19,253	31.39%	6804	18.3%	*p* < 0.001
**CCI**	4	[3, 5]	4	[3, 5]	*p* < 0.001
**HTN**	27,000	27.41%	28,958	29.39%	*p* < 0.001
**Cr**	1.03	±0.79	1.22	±0.75	*p* < 0.001
**HDL**	46.9	±16.93	45.4	±15.58	*p* < 0.001
**LDL**	94	±34.68	83.5	±15.58	*p* < 0.001
**Aspirin**	4886	4.96%	13,871	14.08%	*p* < 0.001
**Statin ***	----		High intensity statins **	10,186 (27.4%)	
			Medium/Low intensity ** statins	26,985 (72.6%)	
			Combination therapy ***	632 (1.7%)	

MI—myocardial infarction; HTN—hypertension; Cr—creatinine; HDL—high-density lipoprotein; LDL—low-density lipoprotein; CCI—Charlson Comorbidity Index. * Rates of high, medium/low and combination therapy are calculated as the rate for all the patients within the statin treatment group (*n* = 37,171). ** High-intensity statins include atorvastatin 40 mg or 80 mg and/or rosuvastatin 20 mg or 40 mg. ** Medium- and low-intensity statins include atorvastatin 20 mg or lower, rosuvastatin 10 mg or lower, simvastatin, pravastatin, lovastatin, fluvastatin, pitavastatin. *** Combination therapy includes a statin of any kind with ezetimibe.

**Table 2 jcm-14-05739-t002:** Comparison of outcomes for both groups with and without chronic statin treatment.

	Without Chronic Statin Treatment (N = 61,331)		With Chronic Statin Treatment (N = 37,171)		
**CVA**	3426	5.58%	6277	16.88%	*p* < 0.001
**MI**	3450	5.62%	6613	17.79%	*p* < 0.001
**365-day mortality**	14,131	23.04%	3805	10.24%	*p* < 0.001
**Total mortality**	42,805	69.79%	19,274	51.85%	*p* < 0.001

MI—myocardial infarction; CVA—cerebrovascular accident.

## Data Availability

The data underlying this article will be shared upon reasonable request to the corresponding author.

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
