# Peer review of "Statin Treatment for Reducing Mortality Risk in Individuals over 75 Years of Age: A Large-Scale Retrospective Analysis"

_jcm, 2025, doi:10.3390/jcm14165739_

Round 1
Reviewer 1 Report
Comments and Suggestions for Authors
This retrospective study investigated the effect of statin treatment on all-cause mortality in individuals aged 75 and older who had no prior history of cardiovascular diseases like angina, myocardial infarction, or stroke. Researchers used real-world data from a large group of outpatients and admitted individuals from the Tel Aviv Sourasky Medical Center. The findings revealed that the risk of all-cause mortality was significantly lower in the group treated with statins compared to the group not receiving statins.
Even though the study is comprehensive, some data points need further clarification. Primarily, the study doesn't mention participants with diagnoses of heart failure or post-ischemic dilated cardiomyopathy. Furthermore, patients with arrhythmias or peripheral artery disease aren't mentioned. Additionally, the study sample ought to be categorized by the medications patients were taking, such as anticoagulants, beta-blockers, or the specific type of antiplatelet drug. The kind of statin used also warrants consideration.
Author Response
Response to Reviewer 1
Comment 1:
The study doesn't mention participants with diagnoses of heart failure or post-ischemic dilated cardiomyopathy. Furthermore, patients with arrhythmias or peripheral artery disease aren't mentioned.
Response: Thank you for your observation. We confirm that patients with peripheral vascular disease and chronic heart failure were excluded using the Charlson Comorbidity Index (CCI), which includes these conditions in its scoring. This is now clarified in the Methods section, under Data Source and variables (lines 110-113, 134-137).
We acknowledge that arrhythmias such as atrial fibrillation were not specifically excluded. This limitation has now been addressed in the Discussion (lines 707 – 712) under the limitations subsection.
Comment 2:
The study sample ought to be categorized by medications such as anticoagulants, beta-blockers, or the specific type of antiplatelet drug. The kind of statin used also warrants consideration.
Response: We agree that a more detailed medication profile would be informative. However, our dataset only included information on aspirin and statin use. We have clarified this in the Methods and Discussion sections (lines 109, 830). Additionally, we categorized statin therapy by intensity and have now included a footnote in Table 1 with the distribution of statin types (lines 503-514).

Reviewer 2 Report
Comments and Suggestions for Authors
I reviewed with interest the manuscript by Noy Nachmias et al. "Statin treatment for reducing mortality risk in individuals over 75 years of age: a large-scale retrospective analysis". In this article, the authors tried to study the possibility of statin therapy to reduce mortality in people over 75 years of age. The analysis was conducted on a large sample of patients seemed to show the effectiveness of statins in reducing overall mortality. However, the study design itself has such significant limitations that the scientific value of the results obtained is significantly reduced.
Comments that arose when reviewing the manuscript:
1. Tables 1 and 2 present data for two groups (patients taking/not taking statins). It is necessary to provide data on statistical differences between the groups for the presented indicators.
2. The authors obtained a paradoxical result - usually the prescription of statins leads to a decrease in mortality due to a decrease in cardiovascular events. In the article, we see the opposite - against the background of taking statins, the number of heart attacks and strokes increased and mortality decreased. The authors tried to somehow explain these paradoxical results, but nevertheless, serious doubts remained about the correctness of collecting retrospective data in this study. In addition, it is obvious that it is necessary to evaluate not only mortality, but also a combined endpoint (non-fatal cardiovascular events + mortality), taking into account such a multidirectional effect of taking statins. 3. In this work, the authors noted a large number of limitations (many non-fatal events could have occurred in other institutions and not been recorded by the researchers, the retrospective nature of the study, the authors could not completely exclude a history of cardiovascular diseases, including peripheral vascular disease and atherosclerotic aortic disease when including patients, etc.), so the conclusions of the study should be less categorical. It should also be noted in the discussion that the use of statins in primary prevention in older people is currently being studied in the STAREE study (refs 1-2, see below).
References:
1. Zoungas S, Curtis A, Spark S, Wolfe R, McNeil JJ, Beilin L, Chong TT, Cloud G, Hopper I, Kost A, Nelson M, Nicholls SJ, Reid CM, Ryan J, Tonkin A, Ward SA, Wierzbicki A; STAREE investigator group. Statins for extension of disability-free survival and primary prevention of cardiovascular events among older people: protocol for a randomized controlled trial in primary care (STAREE trial). BMJ Open. 2023 Apr 3;13(4):e069915. doi: 10.1136/bmjopen-2022-069915.
2. Zoungas S, Moran C, Curtis AJ, Spark S, Flanagan Z, Beilin L, Chong TT, Cloud GC, Hopper I, Kost A, McNeil JJ, Nicholls SJ, Reid CM, Ryan J, Tonkin AM, Ward S, Wierzbicki AS, Wolfe R, Zhou Z, Nelson MR; STAREE investigator group. Baseline Characteristics of Participants in STAREE: A Randomized Trial for Primary Prevention of Cardiovascular Disease Events and Prolongation of Disability-Free Survival in Older People. J Am Heart Assoc. 2024 Nov 19;13(22):e036357. doi: 10.1161/JAHA.124.036357.
Author Response
Response to Reviewer 2
Comment 1:
It is necessary to provide data on statistical differences between the groups for the presented indicators.
Response: We have added p-values for all relevant variables in Tables 1 and 2 to reflect the statistical differences between the statin-treated and untreated groups (lines 503-514 and lines 533-534).
Comment 2:
The authors obtained a paradoxical result—higher rates of MI and CVA despite lower mortality in the statin group. This raises doubts about the accuracy of retrospective data collection. Also, combined endpoints (e.g., MACE) should be evaluated.
Response: We appreciate this important observation. We have elaborated on this paradoxical finding in the Discussion (lines 608-676), noting possible explanations such as higher baseline cardiovascular risk in the statin group, surveillance bias, and limitations in data capture. Due to the nature of our data collection (e.g., national registry for mortality vs. hospital-specific data for events), composite endpoints such as MACE could not be reliably constructed. This is now clearly acknowledged as a limitation (lines 677-679).
Comment 3:
The authors should temper their conclusions due to the many limitations discussed.
Response: We agree and have revised the Discussion and Conclusion sections to ensure that our conclusions are appropriately cautious and clearly framed within the limitations of a retrospective design (lines 697-840).
Comment 4:
Mention the ongoing STAREE trial and include relevant references.
Response: Thank you for the suggestion. We have added a discussion of the STAREE trial and cited both recommended references [Zoungas et al., BMJ Open 2023; J Am Heart Assoc 2024] in the Discussion (lines 684-686) and reference list (references 18, 19).

Reviewer 3 Report
Comments and Suggestions for Authors
This study is not original. The main topic has been published in at least these articles:
https://pmc.ncbi.nlm.nih.gov/articles/PMC8149437/
https://pmc.ncbi.nlm.nih.gov/articles/PMC9572734/
https://www.openhealthgroup.com/publication-library/statin-intake-and-all-cause-mortality-among-older-nursing-home-residents/
https://www.acc.org/Latest-in-Cardiology/Articles/2020/10/01/11/39/Statin-Therapy-in-Older-Adults-for-Primary-Prevention-of-Atherosclerotic-CV-Disease
English is not my native language
Author Response
Response to Reviewer 3
Comment 1:
The study is not original; similar topics have been addressed in prior publications.
Response: We acknowledge that our topic is part of an ongoing scientific discussion. However, our study offers added value by focusing specifically on patients aged ≥75 years, including a substantial subgroup aged ≥85 years, and by using the Charlson Comorbidity Index as a surrogate for ASCVD risk, given the limitations of traditional tools in older adults.
We included the recommended reference (Nowak MM et al., PMC9572734) in our Discussion. We also clarified that in the referenced meta-analysis, only a small fraction of included studies analyzed populations above 75. In addition the referenced article by Anum Saeed et al. (ACC 2020) highlight the gaps that still exist in current guidelines for primary prevention in this population.

Reviewer 4 Report
Comments and Suggestions for Authors
Very interesting work, which clarifies some obscure points in the literature, regarding the use of statins in the elderly in primary prevention.
Unfortunately, the two groups (statin-treated or untreated) differ in age, incidence of high blood pressure, and aspirin use. It is not indicated whether these differences are statistically significant in the description of the population. The authors are asked to indicate this data.
The method of collecting and classifying deaths, causes of death and non-fatal cardiovascular events should be specified Data source and variable
Author Response
Response to Reviewer 4
Comment 1:
The two groups differ in baseline characteristics. Please indicate statistical significance.
Response: Thank you. We have added p-values for all key variables in Table 1 and Table 2 to reflect statistical differences between groups (lines 503-514 and lines 533-534).
Comment 2:
Clarify how deaths, causes of death, and non-fatal cardiovascular events were classified and recorded.
Response: We added a detailed explanation regarding data sources and outcome classification under the Data Source and Variables section (lines 138-141). Specifically, we clarified that all-cause mortality was extracted from the Ministry of Interior national registry, while CVA and MI were identified based on in-hospital troponin elevation and stroke/TIA diagnoses documented at our institution.

Round 2
Reviewer 1 Report
Comments and Suggestions for Authors
No suggestions for the authors
Reviewer 2 Report
Comments and Suggestions for Authors
The authors made significant corrections to the text of the manuscript and answered my questions. I have no other comments.
Reviewer 4 Report
Comments and Suggestions for Authors
Given the revisions of the manuscript.